# Stress Relaxation Properties of Five Orthodontic Aligner Materials: A 14-Day In-Vitro Study

**DOI:** 10.3390/bioengineering9080349

**Published:** 2022-07-28

**Authors:** Paolo Albertini, Valentina Mazzanti, Francesco Mollica, Federica Pellitteri, Mario Palone, Luca Lombardo

**Affiliations:** 1Department of Orthodontics, University of Ferrara, Via Luigi Borsari, 46, 44121 Ferrara, Italy; federica.pellitteri@hotmail.it (F.P.); mario.palone88@gmail.com (M.P.); dot.lulombardo@gmail.com (L.L.); 2Department of Engineering, University of Ferrara, 44121 Ferrara, Italy; valentina.mazzanti@gmail.com (V.M.); albertinie8@gmail.com (F.M.)

**Keywords:** aligner, orthodontics, materials, stress relaxion properties

## Abstract

We aimed to investigate the stress relaxation properties of five different thermoplastic aligner materials subjected to 14 days of constant deflection. Five different thermoplastic aligner materials were selected, whose elastic properties varied: F22 Evoflex, F22 Aligner, Durasoft, Erkoloc-Pro and Duran. The static properties of these materials—in particular, stiffness, stress–strain curve and yield stress—were measured with a three-point bending test. For all the tests that were performed, a minimum of three samples per material were tested. The yield load, yield strength, deformation and particularly the stiffness of each material were found to be similar in the single-layer samples, while the double-layer samples showed far lower stiffness values and were similar one to another. F22 Evoflex and Erkoloc-Pro maintained the highest percentages of stress, 39.2% and 36.9%, respectively, during the 15-day period. Duran and Durasoft obtained the lowest final stress values, 0.5 MPa and 0.4 MPa, respectively, and the lowest percentage of normalized stress, 4.6% and 3.9%, respectively, during the 15-day period. All the materials that we tested showed a rapidity of stress decay during the first few hours of application, before reaching a plateau phase. The F22 Evoflex material showed the greatest level of final stress, with relatively constant stress release during the entire 15-day period. Further research after in vivo aging is necessary in order to study the real aligners’ behavior during orthodontic treatment.

## 1. Introduction

In orthodontics, the need for aesthetic requirements has increased exponentially, due to the diffusion and success of lingual appliances, ceramic brackets and clear aligners [1,2,3]. Recently, aligners have been the subject of several scientific studies; however, the materials used to produce the aligners on the market nowadays vary in terms of thickness and construction.

Align Technology’s (San Jose, CA, USA) early aligners were made of a single-layer hard polyurethane synthesized from methylene-diphenyl-diisocyanate and 1,6-hexanediol. Subsequent material updates were specifically designed to provide improved flexibility, strength and transparency.

Nowadays, glycol-modified polyethylene-terephthalate (PET-G) is the most widely used material for clear aligners, but many other materials are also available. The materials affect the aligner’s mechanical properties and therefore the clinical performance [4,5,6,7,8]. 

Ideally, in order to yield physiological tooth movement, aligners should be able to exert constant light forces throughout their use, but this is difficult to obtain [2]. Orthodontists know the elastic properties of fixed orthodontics archwires, which exert forces that are always proportional to their deflection. Super-elastic copper–nickel–titanium (CuNiTi) wires are even more predictable, as they are able to exert a constant force over a wide deflection range. This property means that the load remains the same, even as the teeth begin to move and align [8,9,10,11,12]. 

Polymers are viscoelastic materials, which have intermediate properties between those of viscous and elastic materials [12]. Even when originally inserted and before any tooth movement is obtained, their behavior under loading might vary significantly over time [13]. A viscoelastic material’s deflection increases with time under constant stress (a phenomenon known as “creep”), while the load necessary to impose a constant deflection decreases over time (a phenomenon known as “stress relaxation”) [14].

Although aligners are subjected to intermittent loads, stress relaxation is relevant, as it entails that even in the case of no tooth movement, the force exerted by the aligner will decrease over time, thus determining a force reduction that may impair tooth movement efficiency. The nature of this reduction, and hence its impact on tooth movement, will be determined by a number of factors, such as the applied load, temperature, the mechanical properties of the aligner material and its geometry. It is therefore important to quantify this decay in order to predict effective tooth movement compared to expectations.

Many studies have been performed in relation to aligner mechanical properties. Shuster et al. [15]. observed a significant increase in aligner stiffness after intraoral wear, due to chewing forces and salivary enzymes. The elastic moduli, hardness and force generated by three different aligner materials were compared by Kohda et al. [16], leading to the discovery of differences in the system of forces that the aligners exert on the teeth.

Only three research works, however, have looked into the stress that aligner materials can produce. Zhang et al. [17] investigated the resistance to traction, stress relaxation and water absorption by various aligner materials after 60 min. Fang et al. [12] studied the stress release at 180 min of five materials at different temperatures. Lombardo et al. [18] investigated the stress release properties of four thermoplastic materials used to produce orthodontic aligners when subjected to 24 consecutive hours of deflection. However, the observation time of these studies was limited and they could not provide a reasonable estimate of the real behavior of aligner materials in the mouth for the duration of treatment.

The goal of this study was to look at the stress relaxation qualities of five different thermoplastic aligner materials that have been deflected continuously for at least 14 days: F22 Evoflex, F22 Aligner, Durasoft, Erokolc-Pro and Duran. The novelty of the present study is represented by the 14-day test, which is the prescription duration for the clinical use of a single aligner. In pursuing this goal, we made use of a well-established practice, known as time–temperature superposition [19], which allows one to test the viscoelastic properties of polymers at relatively short time intervals, but at different temperatures, to infer the mechanical behavior at longer time intervals. In particular, the materials were tested for stress relaxation for one day at two different temperatures, namely 37 °C and 47 °C. The stress data at the higher temperature were used to estimate the material mechanical behavior at 37 °C but for longer times [20]. The null hypothesis of the study was to obtain the same mechanical behavior in all the tested aligner materials.

## 2. Materials and Methods

Five different thermoplastic aligner materials were selected for the present study (Table 1). Three of these materials were 0.75 mm thick single layers, and two were multilayers of thicknesses measuring 1 mm and 1.2 mm, respectively. Because the samples were made of various materials and thicknesses, their elastic properties were different, and their static properties, particularly stiffness, the stress–strain curve and yield stress, were measured with a three-point bending test.

The stress relaxation test was also performed using the three-point bending method, superimposing a constant deflection at the middle section and measuring the force opposed by the material as a function of time. For a more reasonable comparison among the different materials, the deflection value was chosen in such a way that the material reached a quarter of its yield strength, as previously determined through the static testing. This reference value was selected to distinguish the materials’ viscoelastic properties from their elastic ones and to obtain viscoelastic characterization data valid for comparing all the materials, irrespective of their geometry or loading levels.

In fact, for the time–temperature superposition procedure to be valid, the material must be loaded to within its limit of linear viscoelasticity. The stress relaxation data can only be utilized to analyze the stress relaxation response for any deformation or displacement in this scenario. In order to ensure that the linear viscoelasticity limits are not exceeded, the displacement must be small enough. We assumed that 25% of the yield strength of the material would be a compromise value, which was large enough to provide forces that are sufficiently relevant for orthodontic treatment and yet small enough to induce a small strain value that preserved the linear viscoelasticity [20].

For all the tests that were performed, a minimum of three samples per material were tested.

The study design was reviewed and approved by the Ethics Committee (approval number 6/2020).

### 2.1. Yield Strength Testing

Rectangular samples (25 × 50 mm) of each material were obtained from the 125 mm diameter disks provided by the manufacturers. The dimensions and uniformity of each sample were verified with a digital gauge (Vogel, Kevelaer, Germany) at three different points. A three-point bending test was performed on the five materials, applying the ASTM-D790 standard, and we used an INSTRON 4467 (INSTRON, Norwood, MA, USA) dynamometer with a 100 N load cell [21].

After 2 h of conditioning in distilled water at 37 °C to achieve thermal equilibrium, each sample was placed in a bath (20 × 20 × 10 cm) containing distilled water at 37 °C, positioned under the load cell, on a stainless-steel stand with a rectangular base and two equidistant vertical supports, 25 mm apart (the span). In accordance with ASTM-D790, the supports were made with a 1 mm radius of curvature to reduce stress concentrations at the supports.

An immersion heater (Julabo Labortechnik Gmbh, Seelbach, Germany) was positioned in a second water bath of distilled water and connected to the first via intake and outlet pipes through a hydraulic circuit to keep the water temperature at 37 °C. To reduce evaporation, both water baths were covered with plastic film (Figure 1).

A load–deflection test was performed on each sample, with the specimen being deformed at a speed of 100 mm/min to a maximum deflection of 7 mm. An acquisition software program developed within LabView 8.5 was used to record the results (National Instruments Corporation, Austin, TX, USA). After this, a load–deflection curve was created for each sample evaluated using Microsoft Excel (Microsoft Corporation, Redmond, WA, USA). Given the specimen dimensions, i.e., the sample thickness h, its width S and the span L, once the displacement **δ** and the force F were known, as measured by the dynamometer, the following formulae were used to evaluate the maximum strain ε
ε = 6 × hδ/L^2^
and the stress σ
σ = 1.5 × FL/Sh^2^
of each sample.

The subsequent stress relaxation tests were performed by superimposing a constant deflection that caused the sample to reach one fourth of the yield strength of the material, as calculated previously.

### 2.2. Stress Relaxation Testing

Three 25 × 50 mm samples of each material were preconditioned for at least 2 h in distilled water at 37 °C before testing, and then positioned on a stand immersed in distilled water at 37 °C in the same hydraulic circuit as the yield strength test. The Erkoloc-Pro and Durasoft double-layer samples were placed on the stand so that the deflection strip came into contact with their softer layer, which corresponded to the aligner’s internal layer. The established deflection was attained in the first 5 s of the test and remained constant for the next 24 h, during which the load’s relaxation was observed. During the first 30 s, data were collected every 0.5 s, then every second for the next 2 min, and then every 60 s until the end of the test. For each type of material, three tests were conducted to allow for curve comparisons and accurate evaluations of material behavior, as well as to make the analysis statistically valid. For each test, a new sample was used. To compare the stress degradation of each material over the course of a 24-h period, the normalized stress, i.e., the following equation, was used to calculate the percentage of stress decay (normalized stress percent):Normalized Stress % = σ/σmax × 100

The maximum stress reached by each material during the course of the stress relaxation test is σmax, and the stress value measured during the test is σ. This equation was used to calculate the stress decay percentage of each material after 8, 16 and 24 h. All tests were also performed at 47 °C since time–temperature superposition (TTS) is a principle of polymer physics employed to estimate a material’s mechanical behavior at 37 °C but for longer times (Appendix A) [12]. Since a temperature increase determines an acceleration of molecular relaxation phenomena, the relaxation behavior at a certain temperature at a very large loading time can be approximated by the relaxation at a higher temperature but at a much shorter time (Figure 2) [20].

## 3. Results

### 3.1. Yield Strength Testing

The single-layer samples Duran (SCHEU, Iserlohn, Germany), F22 Aligner and F22 Evoflex (Sweden & Martina, Padua, Italy) had similar yield load, yield strength, deformation and, in particular, stiffness, whereas the double-layer samples Erkoloc-Pro (Erkodent, Pfalzgrafenweiler, Germany) and Durasoft (SCHEU) had far lower stiffness values and were similar to each other (Table 2).

Figure 3 shows the stress–strain curves used to calculate the value at which the stress relaxation was tested on each of the four materials, i.e., the deflection at one fourth of their respective yield strength values ranging between 1.04 mm and 1.45 mm (Table 2).

### 3.2. Stress Relaxation Testing

Every specimen generated a different stress relaxation curve during the 24-h period for both temperatures of the test (Figure 4a–e).

An analysis of variance (ANOVA) was conducted to test the stress value of the materials after 14 days, based on the null hypothesis that these values were equal to each other. The analysis resulted in a value of F = 344.87, with *p*-value < 0.001, rejecting the null hypothesis that all materials had the same stress behavior after 14 days. The three specimens of a given material displayed very similar curves, showing excellent repeatability. The curves related to 47 °C were shifted using a suitable shift factor in such a way that the stress relaxation curve at 37 °C could be extended to 15-day periods. For each material, the average curve at 37 °C was then extended with the shifted average curve at 47 °C. The curves during the 15-day period are also compared in Figure 5.

The mean initial stress, final stress and stress decay for each of the five materials are reported in Table 3.

F22 Evoflex polyurethane showed the greatest stress during the 15-day period, with the exception of the first few hours, in which the F22 Aligner generated more absolute stress (Table 3 and Figure 5). Durasoft obtained the lowest stress values, except for the first few hours, in which Erkoloc-Pro registered even lower stress values (Figure 5). Figure 6 and Table 4 show the normalized stress values, i.e., the percentage of stress decay (normalized stress percent), over the 15-day observation period. The lowest percentage of stress decay was observed in Erkoloc-Pro and F22 Evoflex, as opposed to Durasoft and Duran, which showed the greatest decay.

## 4. Discussion

Thickness and material affect performance and differentiate aligners on the market. A previous study [8] analyzed the materials’ stiffness and the minimal force necessary to obtain programmed tooth movements without periodontal support damage; the authors showed a direct correlation between thickness and produced forces.

Recently, another article investigated the stress relaxation properties of different materials on the market when subjected to 24 consecutive hours of applied deflection [18]. The authors found that the single-layer materials showed the greatest values for both absolute stress and stress decay speed; conversely, the double-layer materials showed very constant stress release, but at absolute values up to four times lower than the single-layer samples tested. All the materials showed a higher stress relaxation percentage during the first 8 h under a constant load, and from 16 to 24 h, the percentage of stress relaxation stabilized around a nearly constant plateau.

However, the behavior of aligner materials after 24 h was not investigated in the literature after 24 h; this is a very short time and provides incomplete information for use in the clinical setting [4,8,9,10]. Orthodontic aligners are worn for much longer periods; some authors suggest at least two weeks for biological reasons of tooth movement [21].

An optimal force system is important for an adequate biological response in the periodontal ligament and undermining resorption requires 7–14 days, with the same length of time needed for periodontal ligament (PDL) repair and regeneration [22]. Aligners are subjected to intermittent loads and stress relaxation should not be excessive; otherwise, the aligner force decreases over time, impairing the efficiency of tooth movement.

In this study, the authors used a procedure that is well known in Materials Science to predict the long-term properties of polymeric samples up to 15 days, which exploits the temperature dependence of the relaxation mechanism [20]. This methodology, however, has never been used so far to predict the long-term relaxation behavior of orthodontic aligners and it is the novelty of this study. The materials tested were the same as in the previous study [18] and a new single-layer material, F22 Evoflex, was added to the tests.

The 24-h test curves of the present study report slightly lower values than those of the previously performed study and the repeatability of the tests has increased; these slight variations are probably due to the change in the thermostat, the only device that was changed with respect to the previous protocol [18]. The greatest variations were observed in the Duran material, which recorded values between 12 MPa and 4 MPa in the 24-h stress relaxation tests of the present study, unlike the initial values of 20 MPa, observed in the same test of the previous study. Other articles in the literature are difficult to compare with these experiments, because the materials and protocols are different and a slight variation in an instrument could imply a great variation in the results [12,17,19,23,24]. IiJima et al. [19] chose a different sample geometry, while Zhang et al. [7] and Fang et al. [12] selected different materials. In the present study, the stress relaxation tests confirmed, in all materials, the rapidity of the stress decay during the first few hours of application, before reaching a plateau phase (Figure 4 and Figure 5). This is typical of the phenomenon of polymer stress relaxation [20]. Ideally, aligners should apply light and constant force over time; however, the material should be stiff enough and possess a high enough yield strength in order to provide a force within the elastic range. The effect of these features in the graphs is represented by a horizontal flat curve with a constant force sufficient for tooth movements over time. In the present study, a single-layer material, F22 Evoflex, was also tested, showing high stress values and constant stress release; the final stress was 6.5 MPa and the stress decay was 9.3 MPa (Table 3). The F22 Evoflex material showed the greatest final stress (6.5 MPa) during the 15-day period; Duran and Durasoft obtained the lowest final stress values, 0.5 MPa and 0.4 MPa (Table 3), respectively. The normalized stress curves showed that F22 Evoflex and Erkoloc-Pro maintained the highest percentages of normalized stress, 39.2% and 36.9%, respectively.

Conversely, Durasoft and Duran registered the lowest percentages of normalized stress, 4.6% and 3.9%, respectively. The material thickness greatly affects the force that it generates, but since the thicknesses of the aligners on the market are different, the comparison samples differ with respect to this feature [8]. The aligner materials’ behavior must affect the clinicians’ indications with regard to the number of days needed to wear the same aligner. However, the prescriptions are influenced by other factors, such as the intermittence of use, average patient compliance and tooth movement biology.

The present study has some limitations: the in vitro analysis was performed before the thermoforming phase and the materials were not comparable in terms of composition and thickness [25]. The materials’ performance may change during the wearing period, unlike the in vitro conditions in the simulations. Moreover, despite time–temperature superposition being a well-established procedure in Materials Science, its usage may be questionable in the case of multilayer materials, such as some of the tested aligners. Further research including a finite element study could be performed to clarify the different aligner materials that are best suited to the planned dental movements.

## 5. Conclusions

All materials tested showed the rapidity of the stress decay during the first few hours of application, before reaching a plateau phase. It was noted that the F22 Evoflex material showed the greatest degree of final stress (6.5 MPa), with relatively constant stress release during the 15-day period. F22 Evoflex and Erkoloc-Pro maintained the highest percentages of stress, 39.2% and 36.9%, respectively, during the 15-day period. Duran and Durasoft obtained the lowest final stress values, 0.5 MPa and 0.4 MPa, respectively, and the lowest percentages of normalized stress, 4.6% and 3.9%, respectively, during the 15-day period. The authors conclude that further research after in vivo aging is necessary to study the aligners’ actual behavior during orthodontic treatment.

## Figures and Tables

**Figure 1 bioengineering-09-00349-f001:**
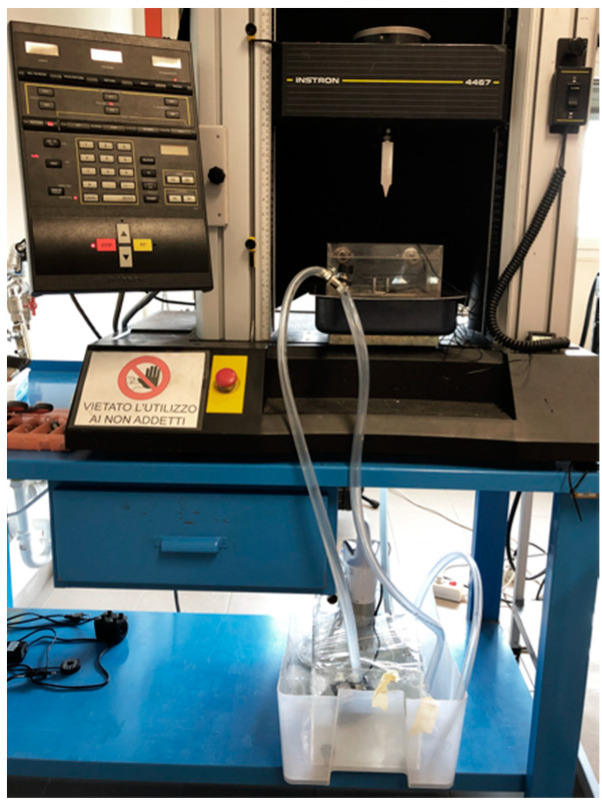
Complete system to perform the tests.

**Figure 2 bioengineering-09-00349-f002:**
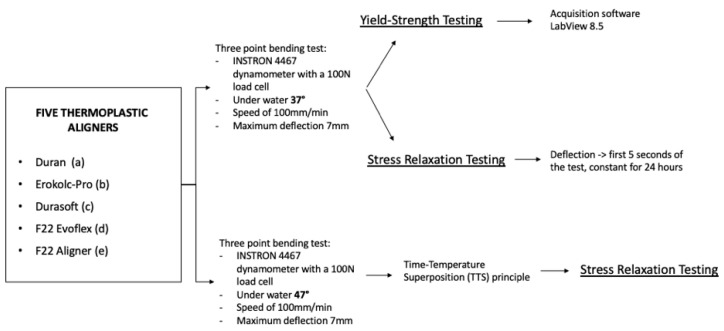
Research flow of the tests employed.

**Figure 3 bioengineering-09-00349-f003:**
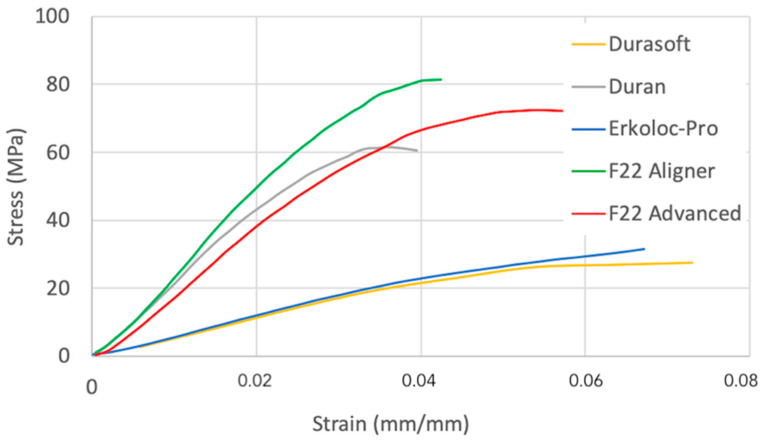
Stress–deformation curve.

**Figure 4 bioengineering-09-00349-f004:**
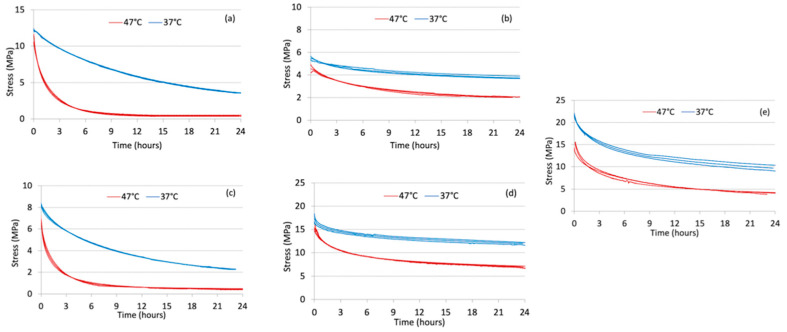
Stress relaxation curve: Duran (**a**), Erkoloc-Pro (**b**), Durasoft (**c**), F22 Evoflex (**d**), F22 Aligner (**e**).

**Figure 5 bioengineering-09-00349-f005:**
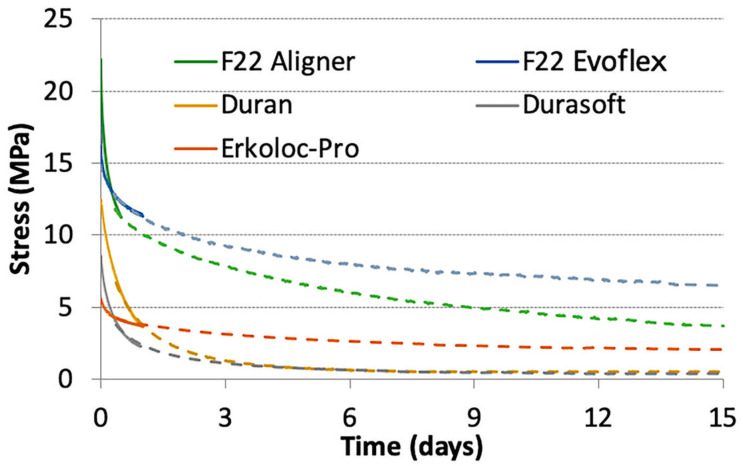
Stress relaxation curves.

**Figure 6 bioengineering-09-00349-f006:**
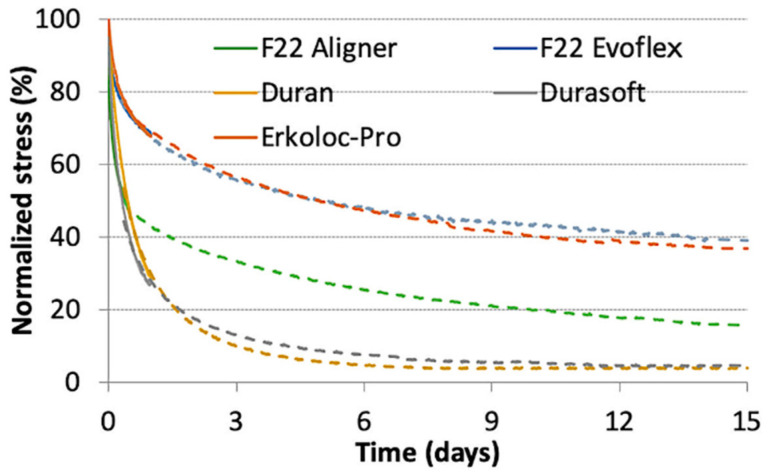
Normalized stress curves.

**Table 1 bioengineering-09-00349-t001:** Materials tested in the study.

Brand Name	Manufacter	Material	Thickness (mm)
F22 Evoflex	Sweden & Martina (Due Carrare, Padua, Italy)	TPU ^a^	0.76
F22 Aligner	Sweden & Martina (Due Carrare, Padua, Italy)	TPU	0.76
Duran	SCHEU (Iserlohn, Germany)	PET-G ^b^	0.75
Erkoloc-Pro	Erkodent (Pfalzgrafenweiler, Germany)	PET-G/TPU	1
Durasoft	SCHEU (Iserlohn, Germany)	TPU/PC ^c^	1.2

^a^ TPU: polyurethane; ^b^ PET-G, polyethylene terephthalate glycol-modified; ^c^ PC, polycarbonate.

**Table 2 bioengineering-09-00349-t002:** Mechanical properties of tested materials (means).

Name	Yield Strength (MPa)	One-Fourth Yield Strength (MPa)	Yield Strain (mm/mm)	Young’s Modulus (MPa)	Deflection at One-Fourth Yield Strength (mm)
F22 Aligner	81.36	20.34	0.0424	2770	1.26
F22 Evoflex	72.35	18.08	0.0537	2104	1.21
Duran	61.63	15.41	0.0371	2366	1.04
Erkoloc-Pro	31.53	7.88	0.0672	597	1.45
Durasoft	27.57	6.89	0.0730	583	1.1

**Table 3 bioengineering-09-00349-t003:** Mean initial stress, final stress and stress decay (means).

Name	σ_1_ (MPa)	σ_2_ (MPa)	Stress Decay σ_1_–σ_2_ (MPa)
F22 Aligner	22.2	3.7	18.5
F22 Evoflex	15.8	6.5	9.3
Duran	12	0.5	11.5
Erkoloc-Pro	5.2	2.1	3.1
Durasoft	7.8	0.4	7.4

**Table 4 bioengineering-09-00349-t004:** Normalized stress relaxation (means).

	Time (Days)			
	0	5	10	15
Normalized stress (%)				
F22 Aligner	100	27.4	19.9	15.7
F22 Evoflex	100	50.3	43.7	39.2
Duran	100	5.6	3.9	3.9
Erkoloc-Pro	100	49.8	40.2	36.9
Durasoft	100	8.6	5.3	4.6
Stress decay (%)				
F22 Aligner	0	72.6	80.1	84.3
F22 Evoflex	0	49.7	56.3	60.8
Duran	0	94.4	96.1	96.1
Erkoloc-Pro	0	51.2	59.8	63.1
Durasoft	0	91.4	94.7	95.4

## Data Availability

Not applicable.

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
