# Peer review of "Stress Relaxation Properties of Five Orthodontic Aligner Materials: A 14-Day In-Vitro Study"

_bioengineering, 2022, doi:10.3390/bioengineering9080349_

Round 1
Reviewer 1 Report
1. Remove the yellow highlight on First Author.
2. In lines 6-18, the email of the authors should be written in black color without an underline.
3. “Aligner” in the keywords section should be using the lowercase font.
4. What is the novelty of the present study? In vitro study of Aligner, materials have been widely studied in the past. There is nothing really new brings in the present manuscript that makes it lack scientific contribution. The authors need to highlight their novelty in the introduction section.
5. In line 57-58, to support the explanation of polymer material's behavior, it is recommended to adopt additional reference published by MDPI as follow: Computational Contact Pressure Prediction of CoCrMo, SS 316L and Ti6Al4V Femoral Head against UHMWPE Acetabular Cup under Gait Cycle. J. Funct. Biomater. 2022, 13, 64. https://doi.org/10.3390/jfb13020064
6. Please make the paragraph from 3 sentences or more that consist of one topic sentence and followed by a supporting sentence to make a solid explanation in the paragraph. The author sometimes constructs a paragraph with only two sentences, for example in lines 47-49. Revise it.
7. Research flow needs to be explained as a form of illustration to make the reader easier to understand in the materials and methods section.
8. The authors need to give detailed information including manufacturer and region regarding tools specification used for experimental testing.
9. What is the basis for conducting only 14 days of In Vitro study? It is too short which would be leading to biased results. A longer time is needed for a more in-depth investigation of stress relaxation of the aligner materials.
10. The discussion needs more extended for a more comprehensive explanation. The discussion on the side of biocompatible and the possibility of in silico study (using finite element model) should be added. Recommended reference from comments number 5 would be used to support this explanation.
11. Accuracy and error in texting should be explicitly explained in the manuscript as information due to different results in further studies.
12. Mechanical properties of texted materials should be given for better understanding.
13. The present results need to be discussed and compared with previous similar studies.
14. If there are any other limitations in the present study, please mention them.
15. Further research needs to be mentioned in the conclusion section.
16. References need to enrich with literature published in the previous 5 years to show there since it only used 14 references. It is encouraged to use references published by MDPI.
Author Response
Dear Editor,
we answer questions from Reviewers point by point.
Reviewer 1:
- Remove the yellow highlight on First Author.
The yellow highlight has been removed
- In lines 6-18, the email of the authors should be written in black color without an underline.
The font from lines 6-18 has been corrected
- “Aligner” in the keywords section should be using the lowercase font.
“aligner” as keyword has been corrected
- What is the novelty of the present study? In vitro study of Aligner, materials have been widely studied in the past. There is nothing really new brings in the present manuscript that makes it lack scientific contribution. The authors need to highlight their novelty in the introduction section.
“The novelty of the present study represents the 14-day test, which are the prescription days for clinical use of the single aligner.” Was added to the introduction section.
- In line 57-58, to support the explanation of polymer material's behavior, it is recommended to adopt additional reference published by MDPI as follow: Computational Contact Pressure Prediction of CoCrMo, SS 316L and Ti6Al4V Femoral Head against UHMWPE Acetabular Cup under Gait Cycle. J. Funct. Biomater. 2022, 13, 64. https://doi.org/10.3390/jfb13020064
The reference has been added.
- Please make the paragraph from 3 sentences or more that consist of one topic sentence and followed by a supporting sentence to make a solid explanation in the paragraph. The author sometimes constructs a paragraph with only two sentences, for example in lines 47-49. Revise it.
The paragraph has been revised and corrected.
- Research flow needs to be explained as a form of illustration to make the reader easier to understand in the materials and methods section.
Illustration flow of materials and methods has been added.
- The authors need to give detailed information including manufacturer and region regarding tools specification used for experimental testing.
The information of manufacturer and region has been added to every specific tool used in the experiments.
- What is the basis for conducting only 14 days of In Vitro study? It is too short which would be leading to biased results. A longer time is needed for a more in-depth investigation of stress relaxation of the aligner materials.
The basis of conduction only 14 days in vitro study is based on the treatment prescription: aligners are worn for a maximum of 14 days per aligner, then new aligners are changed and worn for another 14 days, and so on.
- The discussion needs more extended for a more comprehensive explanation. The discussion on the side of biocompatible and the possibility of in silico study (using finite element model) should be added. Recommended reference from comments number 5 would be used to support this explanation.
The discussion was improved as requested.
- Accuracy and error in texting should be explicitly explained in the manuscript as information due to different results in further studies.
An ANOVA analysis of variance was conducted to test the stress value of the materials after 14 days, based on the null hypothesis that these values were equal to each other. The analysis resulted in a value of F = 344.87, with p-value <0.001, rejecting the null hypothesis that all materials had the same stress behavior after 14 days.
- Mechanical properties of texted materials should be given for better understanding.
The mechanical properties of the tested materials have been added (Table 2).
- The present results need to be discussed and compared with previous similar studies.
The discussion has been improved with recent similar studies
- If there are any other limitations in the present study, please mention them.
The limitation section was improved.
- Further research needs to be mentioned in the conclusion section.
Further research possibilities were added.
- References need to enrich with literature published in the previous 5 years to show there since it only used 14 references. It is encouraged to use references published by MDPI.
The references have been enriched with previous 5 years studies.
Thank You for Your interest in the study and for Your comments.

Reviewer 2 Report
Dear Authors,
The aim of this study was to investigate the stress relaxation properties of five different thermoplastic aligner materials subjected to 14 days of constant deflection. While the topic is fitting to the journal scope, some concerns were raised to publish as a scientific paper. Revise the manuscript by following comments.
Major points
Introduction
Some related literatures are lacking. Make sure the following literatures.
Inoue S, et al., Orthodontic Aligner Incorporating Eucommia ulmoides Exerts Low Continuous Force: In Vitro Study. Materials 13(18):4085, 2020.
Inoue S, et al., Influence of constant strain on the elasticity of thermoplastic orthodontic materials. Dent Mater J 39(3):415-421, 2020.
Table 1
The authors used five different thermoplastic materials with different thickness. Is there any effect for results?
Figure 1 and Figure 2 are not meaningful. They could be removed.
Minor points
Section 2.1
"the displacement xx" might be typo.
Equation number was same format with the reference style. Is it correct?
Superscript in the equations should be reflected.
Section 2.2
"372" is correct?
Figure 4
Numbers in the horizontal axis should be modified. E.g. "0,08" -> "0.08".
Figure 5
"A" and "(a)" were duplicated. Unified to be one. Same for others.
Table 3
"Mean Initial Stress xxx" should be modified to "Mean initial stress xxx". Same for Table 4 legend.
Supplemental
Figure A1, A2, and A3 should be rephrased to Figure S1, S2, and S3 as supplementals.
Figure A1
There was no meaning for colored lines.
Figure A2
There was no explanation for a line and a dotted line.
Figure A3
There was no explanation about three lines.
Author Response
Dear Editor,
we answer questions from Reviewers point by point.
Reviewer 2:
Dear Authors,
The aim of this study was to investigate the stress relaxation properties of five different thermoplastic aligner materials subjected to 14 days of constant deflection. While the topic is fitting to the journal scope, some concerns were raised to publish as a scientific paper. Revise the manuscript by following comments.
Major points
- Introduction: Some related literatures are lacking. Make sure the following literatures:
- Inoue S, et al., Orthodontic Aligner Incorporating Eucommia ulmoides Exerts Low Continuous Force: In Vitro Study. Materials 13(18):4085, 2020.
- Inoue S, et al., Influence of constant strain on the elasticity of thermoplastic orthodontic materials. Dent Mater J 39(3):415-421, 2020.
The suggested references have been added.
- Table 1: The authors used five different thermoplastic materials with different thickness. Is there any effect for results?
Yes, this aspect has been described in the limitations section.
- Figure 1 and Figure 2 are not meaningful. They could be removed.
Figures 1 and 2 are included in the text to show the load cell, which is ideal for testing aligners. However, if the reviewer feels they are not needed, they can be removed from the article.
Minor points
- Section 2.1
- "the displacement xx" might be typo.
The d has been corrected.
- Equation number was same format with the reference style. Is it correct?
The equation number has been removed for better comprehension.
- Superscript in the equations should be reflected.
Superscript in the equations has been corrected.
- Section 2.2
- "372" is correct?
The temperature number has been corrected.
- Figure 4
- Numbers in the horizontal axis should be modified. E.g. "0,08" -> "0.08".
The numbers have been modified.
- Figure 5
- "A" and "(a)" were duplicated. Unified to be one. Same for others.
The letters have been modified.
- Table 3
- "Mean Initial Stress xxx" should be modified to "Mean initial stress xxx". Same for Table 4 legend.
Table 3 legend has been modified.
- Supplemental
- Figure A1, A2, and A3 should be rephrased to Figure S1, S2, and S3 as supplementals.
The figures have been renamed.
- Figure A1
- There was no meaning for colored lines.
The colored lines have been used in order to better distinguish the tests curves from the graph lines.
- Figure A2
- There was no explanation for a line and a dotted line.
A better explanation of the graph has been added.
- Figure A3
- There was no explanation about three lines.
A better explanation of the graph has been added.
Thank You for Your interest in the study and for Your comments.

Reviewer 3 Report
Dear all, here are my concerns regarding the manuscript entitled: “Stress Relaxation Properties of Five Orthodontic Aligner Materials: A 14-Day In Vitro Study”
I recommend the authors to remove the heading from the Abstract following the journal guidelines.
I suggest the authors to add the term “Orthodontics” to the keywords.
The introduction and rational look good. I would suggest the use of a null hypothesis in complement to the aim sentence. Moreover, I would say which are the 5 different thermoplastic aligners and not only state that you intend to evaluate 5.
Regarding the sentence: “For all the tests that were performed, a minimum of three samples per material were tested.” The final sample size was just n=3, it that it? Why did the author conduct the experiment in a so small sample size and how did you reach this sample size.
In Table 2, 3 and 4. Are those values an average?
I advise the authors to debate the study strengths, internal validity, generalization of the results and further research perspectives in the Discussion.
Author Response
Dear Editor,
we answer questions from Reviewers point by point.
Reviewer 3:
Dear all, here are my concerns regarding the manuscript entitled: “Stress Relaxation Properties of Five Orthodontic Aligner Materials: A 14-Day In Vitro Study”
- I recommend the authors to remove the heading from the Abstract following the journal guidelines.
The abstract has been corrected.
- I suggest the authors to add the term “Orthodontics” to the keywords.
The suggested word has been added to the key words.
- The introduction and rational look good. I would suggest the use of a null hypothesis in complement to the aim sentence. Moreover, I would say which are the 5 different thermoplastic aligners and not only state that you intend to evaluate 5.
The null hypothesis has been added. The five thermoplastic aligners have been added.
- Regarding the sentence: “For all the tests that were performed, a minimum of three samples per material were tested.” The final sample size was just n=3, it that it? Why did the author conduct the experiment in a so small sample size and how did you reach this sample size.
The same sample size and number of tests is based on similar previous studies cited in the manuscript. Moreover, as shown in the figure 5 the three repeated curves are almost perfectly superimposed, demonstrating the great reproducibility.
- In Table 2, 3 and 4. Are those values an average?
Yes, it has been added.
- I advise the authors to debate the study strengths, internal validity, generalization of the results and further research perspectives in the Discussion.
The discussion has been improved as requested and further research perspectives has been added.
Thank You for Your interest in the study and for Your comments.
Round 2
Reviewer 1 Report
The present form is recommended for publication.
Author Response
Thank You very much.
The Authors
Reviewer 2 Report
Dear Authors,
The manuscript was mostly revised according to the reviewer's comments. Some minor concerns were still remained. Revise the manuscript by following comments.
Minor points
Figure 1 and Figure 2 could be removed. Revise the related sentences.
Table 2 legend must be located on the top of table.
Figure S3
"blu line" is typo. Revise it.
Author Response
Dear Editor,
we answer questions from Reviewer point by point.
Reviewer 2:
Minor points
- Figure 1 and Figure 2 could be removed. Revise the related sentences.
Figure 1 and 2 has been removed and the figures captions has been corrected.
- Table 2 legend must be located on the top of table.
Table 2 legend has been located on the top of the table
- Figure S3: "blu line" is typo. Revise it.
Figure S3 caption has been corrected.
Thank You for Your interest in the study and for Your comments.
The Authors
Reviewer 3 Report
Dear authors, I have no more concerns. Thank you.
Author Response
Thank You very much.
The Authors